# Examining Chlorophyll Extraction Methods in Sesame Genotypes: Uncovering Leaf Coloration Effects and Anatomy Variations

**DOI:** 10.3390/plants13121589

**Published:** 2024-06-07

**Authors:** Muez Berhe, Jun You, Komivi Dossa, Donghua Li, Rong Zhou, Yanxin Zhang, Linhai Wang

**Affiliations:** 1Oil Crops Research Institute of the Chinese Academy of Agricultural Sciences, Key Laboratory of Biology and Genetic Improvement of Oil Crops, Ministry of Agriculture and Rural Afairs, No. 2 Xudong 2nd Road, Wuhan 430062, China; muez.sbn@gmail.com (M.B.);; 2Tigray Agricultural Research Institute, Humera Agricultural Research Center, Mekele P.O. Box 62, Ethiopia; 3CIRAD, UMR AGAP Institut, 97170 Petit Bourg, Guadeloupe, France; komivi.dossa@cirad.fr; 4UMR AGAP Institut, Univ Montpellier, CIRAD, INRAE, Institut Agro, F-34398 Montpellier, France

**Keywords:** carotenoids content, chloroplast ultrastructure, leaf disk positions, *Sesamum indicum*, SPAD value index

## Abstract

This study focuses on optimizing chlorophyll extraction techniques, in which leaf discs are cut from places on the leaf blade to enhance chlorophyll concentration in sesame (*Sesamum indicum* L.) leaves. Thirty sesame genotypes, categorized into light green (LG), middle green (MG), and deep green (DG) pigment groups based on leaf coloration, were selected from a larger pool of field-grown accessions. The investigation involved determining optimal Soil Plant Analysis Development (SPAD) value index measurements, quantifying pigment concentrations, exploring extraction solvents, and selecting suitable leaf disk positions. Significant variations in chlorophyll content were observed across genotypes, greenness categories, and leaf disk positions. The categorization of genotypes into DG, MG, and LG groups revealed a correlation between leaf appearance and chlorophyll content. The study highlighted a consistent relationship between carotenoids and chlorophyll, indicating their role in adaptation to warm environments. An examination of leaf disk positions revealed a significant chlorophyll gradient along the leaf blade, emphasizing the need for standardized protocols. Chlorophyll extraction experiments identified DMSO and 96% ethanol, particularly in those incubated for 10 min at 85 °C, as effective choices. This recommendation considers factors like cost-effectiveness, time efficiency, safety, and environmental regulations, ensuring consistent and simplified extraction processes. For higher chlorophyll extraction, focusing on leaf tips and the 75% localization along the sesame leaf blade is suggested, as this consistently yields increased chlorophyll content. Furthermore, our examination revealed significant anatomical variations in the internal structure of the mesophyll tissue leaves between deep green and light green sesame plants, primarily linked to chloroplast density and pigment-producing structures. Our findings, therefore, provide insightful knowledge of chlorophyll gradients and encourage the use of standardized protocols that enable researchers to refine their experimental designs for precise and comparable chlorophyll measurements. The recommended solvent choices ensure reliable outcomes in plant physiology, ecology, and environmental studies.

## 1. Introduction

Chlorophyll, the green pigment found in plant leaves, plays a fundamental role in photosynthesis, the process by which plants convert light energy into chemical energy, ultimately sustaining life on Earth. It serves as a crucial indicator of a plant’s photosynthetic vitality and overall health. Measuring chlorophyll concentration in plants is crucial in assessing a plant’s photosynthetic efficiency, overall health, and response to environmental stressors. In terms of its structure, chlorophyll consists of a tetra pyrrole ring housing a central magnesium ion, accompanied by an elongated hydrophobic phytol chain [1]. Its presence extends across a wide spectrum of organisms, encompassing higher plants, ferns, mosses, green algae, and specific prokaryotic entities like Prochloron varieties inhabiting plants and algae [2,3].

In higher plants, chlorophyll manifests primarily as chlorophyll a (Chla), serving as the principal pigment, and chlorophyll b (Chlb), operating as a supplementary pigment. These chlorophyll variants coexist in a typical chlorophyll ratio (Chla/b) of roughly 3 to 1, although this proportion can be subject to fluctuations influenced by growth conditions and environmental factors [3]. The primary differentiation between these two chlorophyll types is the presence of a methyl group in Chla, which is replaced by a formyl group in Chlb [3,4]. Chlorophyll predominantly absorbs light within the red (650–700 nm) and blue-violet (400–500 nm) regions of the visible spectrum and, interestingly, it exhibits a unique property of reflecting green light (~550 nm), which imparts the characteristic green coloration to chlorophyll [1,3,4,5].

Sesame, a valuable oilseed crop, is highly regarded for its oil-rich seeds, finding applications in cooking, cosmetics, and traditional medicine [6,7]. This crop is esteemed for its resilience to varying climates, impressive oil content, and exceptional antioxidant properties [8]. Sesame serves as a vital source of premium-quality edible oil and protein-rich food. Sesame seeds typically contain oil in the range of 50–60%, and this oil boasts a significant proportion of natural antioxidants, such as sesamolin, sesamin, and sesamol; these antioxidants contribute to the extended shelf life and stability of sesame oil [6,9]. Studies have revealed that sesame seeds are rich in protein, which comprises approximately 19–25% of their composition [6]. Moreover, sesame seeds are abundant in essential minerals like iron, magnesium, copper, and calcium, along with vitamins B1 and E, as well as phytosterols; these nutritional components were reported to play a role in reducing blood cholesterol levels [10]. Additionally, sesame seeds encompass all the essential amino acids and fatty acids, further enhancing their nutritional value [11]. Sesame, therefore, emerges as a versatile and nutritious crop with immense potential for various applications, from culinary to health-enhancing products. In this case, understanding and optimizing chlorophyll extraction from sesame leaves are is particular importance because chlorophyll content can provide insights into the crop’s health, growth, and potential yield.

Accurately quantifying chlorophyll concentration demands the utilization of effective extraction techniques. Traditionally, wet chemical methods have entailed dissolving chlorophyll in a solvent, followed by assessing the absorbance of the chlorophyll solution via spectrophotometry. Concentration values are then derived using well-established equations. Previous investigations have explored a range of methods to evaluate chlorophyll content in higher plant leaves, encompassing non-destructive chlorophyll meters [12], fluorometry [13], photo-acoustic spectroscopy [14], chromatographic approaches [15], and spectrophotometry [4,16,17,18]. Numerous studies have demonstrated that chlorophylls, which are lipid-soluble compounds present in plant tissues, can be extracted using water-miscible organic solvents. These solvents include acetone, pyridine, methanol, ethanol, diethyl ether, DMF, and DMSO, all of which have the ability to absorb water [4,16,19,20,21,22,23,24,25,26,27].

Despite extensive experimentation dedicated to chlorophyll extraction and quantification, no single method has been identified that simultaneously offers simplicity, widespread applicability, ease of reproducibility, and high sensitivity [25]. The presence of numerous variable conditions affecting chlorophyll pigments, their extractability, and their reactions complicates the task of selecting a single extraction procedure that can accurately estimate all the green components present in various plants and plant-derived products. Different researchers have reported varying degrees of effectiveness with different extraction procedures. The acetone method [4] has been widely accepted for chlorophyll determination due to its simplicity, sensitivity, and safety compared to the DMSO and DMF methods [28,29]. However, the acetone method involves grinding plant tissue in acetone, followed by centrifugation, which can be time-consuming for large sample numbers [24]. In contrast, the DMSO method is simpler and faster as it eliminates the need for grinding and centrifugation [12], offering similar efficiency to acetone with superior chlorophyll stability [24]. Nevertheless, DMSO and DMF are associated with toxicity and unpleasant odors during extraction [29], and DMSO extraction is sensitive to temperature fluctuations [30]. When comparing acetone and ethanol, both considered less harmful solvents, 96% ethanol has been reported as an efficient extraction solvent compared to 90% acetone [31].

Given the economic significance of sesame and the importance of chlorophyll as an indicator of plant health and productivity, there is a pressing need to optimize chlorophyll extraction techniques to ensure accurate and efficient measurements. This research aims to address these challenges by exploring innovative methods and leaf disk positions to enhance the precision and reliability of chlorophyll extraction from sesame leaves. The findings of this study can have broader implications for agriculture and plant biology. By improving chlorophyll measurement techniques, researchers, agronomists, and farmers can better monitor and manage the health and growth of sesame crops, leading to increased yields and sustainable agricultural practices.

## 2. Results

### 2.1. SPAD Value Index (SVI) Variation within Sesame Genotypes

The ANOVA analysis for the SPAD Value Index (SVI) reveals significant effects attributed to the genotype variation (F29, 420 = 25.41; *p* < 0.001) on the relative chlorophyll concentration of sesame leaves. The results presented in Figure 1A underscore the significant variation in chlorophyll content among sesame genotypes. Among the evaluated sesame genotypes, a wide range of chlorophyll content was observed. Notably, DG5 and DG3 exhibited the highest levels of chlorophyll content, signifying their exceptional chlorophyll production capabilities. DG5, in particular, stands out, with remarkably high chlorophyll content. Moving to the medium green category, we continue to observe variability in chlorophyll content. MG5 and MG7 fall within the intermediate range, indicating moderate chlorophyll content. MG6 and MG3 are also categorized here, with slightly lower chlorophyll content compared to MG5 and MG7. Conversely, LG6, LG9, and LG7 are the genotypes that exhibit the lowest chlorophyll content.

### 2.2. SPAD Value Index (SVI) Variation among Different Sesame Greenness Categories

The SPAD Value Index (SVI) ANOVA analysis reveals significant effects of the greenness category (F2, 447 = 191.53; *p* < 0.001) on the relative chlorophyll concentration in sesame leaves. Categorizing sesame genotypes into DG, MG, and LG groups elucidates the diverse chlorophyll content among genotypes, emphasizing substantial variations, irrespective of genotype differences. In Figure 2B, the distinct chlorophyll difference among DG, MG, and LG categories is evident. DG plants exhibit the highest chlorophyll content, with a mean SVI of 51.052 and a range from 29.00 to 59.70 (Figure 1B), significantly differing from MG and LG. MG, with a mean chlorophyll content of 45.839 and a range from 31.100 to 61.000 (Figure 2B), shows a lower chlorophyll content than DG but a significantly higher content than LG. LG, with the lowest mean chlorophyll content at 38.15 and a range from 20.30 to 55.40 (Figure 2B), indicating a significantly lower chlorophyll content compared to DG and MG. These findings emphasize the substantial impact of greenness categories on chlorophyll content, providing insights into the relative chlorophyll concentrations of sesame genotypes. The categorization based on greenness proves informative, offering a clearer understanding of the inherent chlorophyll variations among sesame plants.

### 2.3. Variation in SPAD Value Index (SVI) across Different Leaf Disk Localizations along the Leaf Blade

The ANOVA analysis for the SPAD Value Index (SVI) reveals significant effects, attributed to the leaf disk position (F4, 445 = 19.46; *p* < 0.001), on the relative chlorophyll concentration of sesame leaves. The study reveals notable variations in chlorophyll concentration across different localizations of sesame leaves (Figure 2). The leaf tip exhibits the highest chlorophyll content, with a mean SVI of 49.00 and a range of 34.40 to 61.00, significantly differing from other positions. At the 75.00% localizations, the leaf disk shows an intermediate chlorophyll level (47.00) that is not significantly different from the tip but distinct from the base. The middle localizations, with a mean SVI of 45.45, display moderate chlorophyll content, differing significantly from the base. The 25.00% localizations present with intermediate chlorophyll levels (43.40), which are significant compared to other positions except the base. The leaf base has the lowest mean SVI (40.25) and significantly differs from all other positions, indicating markedly lower chlorophyll content. These findings emphasize a clear chlorophyll gradient along the leaf blade, with the tip demonstrating superior content compared to other positions, particularly the base.

### 2.4. Comparison of Extraction Method for Optimal Chlorophyll Extraction

The ANOVA analysis unequivocally identified significant variations in chlorophyll concentrations (Chla, Chlb, and Chla+b) across various extraction methods (F5, 36 = 17.76, *p* < 0.001; F5, 36 = 5.01, *p* < 0.001; F5, 36 = 21.71, *p* < 0.001, respectively). However, no significant differences were observed in Chla/b (F5, 36 = 0.42, *p* = 0.831). In-depth mean comparisons revealed that DMSO incubation at 65 °C, 96% ethanol incubation at 85 °C, and boiling leaves in 96% ethanol at 85 °C yielded Chla levels that were significantly higher than other methods, although they did not significantly differ from each other (Figure 3A–D). Notably, DMSO exhibited the highest Chla content (mean: 27.22), emphasizing its efficacy. Conversely, 96% ethanol at 65 °C consistently yielded the lowest Chla content. These findings underscore the crucial influence of extraction methods on Chla levels and highlight DMSO’s prominence.

Additionally, DMSO and 96% ethanol at 85 °C consistently produced higher Chlb and total chlorophyll content, while 96% ethanol at boiling point and 96% ethanol at 40 °C for 24 h displayed intermediate levels. Solvents 75% ethanol at 650 °C and 96% ethanol at 65 °C consistently resulted in the lowest Chlb and total chlorophyll content. The Chla/b ratio exhibited subtle variations among extraction methods, ranging from a mean of 2.910 to 4.81. These findings emphasize the critical need for methodological precision in chlorophyll analysis, offering insights for optimizing protocols to ensure accurate and reliable measurements in plant samples.

#### Efficiency of DMSO and 96% Ethanol as Optimal Chlorophyll Extraction Solvents

Both DMSO and 96% ethanol emerged as highly efficient chlorophyll extraction solvents in this study, demonstrating a comparable performance. While DMSO extracted a slightly higher total Chla content (33.59) compared to 96% ethanol (29.392) across all three extraction frequencies, statistical analysis did not detect a significant variation between the two solvents (Table 1). Notably, the ANOVA results highlighted a significant difference in Chla extraction efficiency among the different extraction frequencies within each solvent. The initial extraction proved most effective for both solvents, contributing approximately 81% of the total Chla (Table 1). Subsequent extractions, though yielding similar amounts of Chla for DMSO (17.35%) and 96% ethanol (16.94%), were significantly lower than their corresponding first extracts. The third extraction was the least efficient for both solvents, resulting in very low Chla content (DMSO at 1.62% and 96% ethanol at 2.03%) (Table 1).

Similar trends were observed for Chlb extraction, with significant differences among extraction frequencies within each solvent but no significant variation between DMSO and 96% ethanol. The initial extraction was most efficient, constituting the majority of Chlb content (approximately 85% and 77% for 96% ethanol and DMSO, respectively) (Table 1). Subsequent extractions showed diminishing yields, with the third extraction being the least efficient.

Both solvents efficiently extracted total Chla+b, with no significant difference between them. The first extraction dominated in terms of efficiency, contributing the majority of total chlorophyll content (around 82% for 96% ethanol and 80% for DMSO). The second extraction extracted a lower percentage (around 15–18%), and the third extraction contributed only a negligible amount (Table 1). For Chla/b extraction, both solvents performed efficiently, with no significant difference. Similar to Chla and Chlb, the first extraction was the most efficient, resulting in the highest Chla/b content, while subsequent extractions produced lower chlorophyll ratios. The findings underscore the robust performance of both DMSO and 96% ethanol, emphasizing the significance of the initial extraction in maximizing chlorophyll yield.

### 2.5. Quantification of Chlorophyll Concentration Variation in Sesame Leaves

The overall ANOVA results of an analysis for chlorophyll concentration in sesame genotypes based on both genotype variations and leaf disk positions demonstrated that both genotype (F29, 180 = 15.65, *p* < 0.001) and leaf disk localization (F2, 180 = 11.65, *p* < 0.001) have significant effects on chlorophyll concentration in sesame genotypes using 96% ethanol incubated at 85 °C for 10 min. The interaction between these factors (F58, 180 = 0.99, *p* = 0.50) however, does not appear to contribute significantly to the variation in chlorophyll concentration. The variation in genotype has a significant effect on chlorophyll concentration, which suggested that there are significant differences in chlorophyll concentration among the different sesame genotypes being studied. Similarly, the chlorophyll concentration varies significantly depending on the position at which leaf disks are taken within the leaf blade (Figure 4A–G).

#### 2.5.1. Variation in Chlorophyll Concentration across Sesame Genotypes and Leaf Greenness Categories

The ANOVA analysis in this study unveiled significant diversity in Chla concentration among various sesame genotypes (F29, 240 = 14.4, *p* < 0.001), and different greenness categories (F2, 267 = 80.34). DG5 (mean = 28.55) and DG10 (mean = 27.42) stood out as exceptional genotypes with the highest Chla content, surpassing other genotypes. MG6 (24.22) and DG2 (22.19) displayed intermediate Chla concentrations, while LG7 (13.32), LG10 (13.860) and MG10 (13.45) exhibited the lowest Chla levels (Figure 4A). Further insights from Figure 4B emphasized the link between Chla concentration and greenness categories. DG genotypes demonstrated significantly a higher mean Chla concentration (25.30) compared to the MG (18.279) and LG (17.529) categories, with no statistically significant differences between MG and LG.

Chl_b_ concentration exhibited significant variation among genotypes (F24, 240 = 13.47, *p* < 0.001) and greenness categories (F2. 267 = 67.28, *p* < 0.001) (Figure 4C). Several genotypes, including DG9, DG10, DG5, DG7, DG6, and MG6, showcased the highest mean Chlb concentrations (11.49 to 11.94), indicating a robust capacity for Chl_b_ production. Genotypes DG4 and DG3 displayed intermediate Chlb concentrations (10.75 to 10.77), demonstrating noteworthy Chl_b_ content. Another group of genotypes exhibited lower mean Chl_b_ concentrations (5.71 to 10.00), indicating relatively lower Chl_b_ content. Similar to Chl_a_ trends, the DG category (mean = 10.85) displayed a significantly higher mean Chlb concentration compared to MG (mean = 8.03) and LG (mean = 7.55) categories, reaffirming the association between deep green characteristics and higher Chlb content in DG genotypes. These findings elucidate the intricate relationship between genotype, greenness category, and chlorophyll composition in sesame plants, offering valuable insights into the factors influencing chlorophyll variability (Figure 4D).

Regarding the total chlorophyll content, the ANOVA analyses underscore substantial variations in total chlorophyll content among sesame genotypes (F29, 240 = 14.21, *p* < 0.001) and greenness categories (F2, 267 = 76.95, *p* < 0.001), revealing diversity in chlorophyll production. Figure 4E depicts distinct total chlorophyll levels among genotypes, with DG5 leading at 40.35, followed by DG10, DG6, DG9, DG7, and DG4, with elevated content. Genotypes MG6 and DG3 fall within an intermediate range, while others show lower chlorophyll content. Greenness categories mirror the patterns seen in Chl_a_ and Chl_b_ (Figure 4F).

Figure 4G illustrates significant variability in carotenoid content among genotypes (F29, 240 = 14.93, *p* < 0.001). DG5, DG4, DG6, DG10, MG6, and DG3 exhibit noteworthy carotenoid levels, contrasting with LG9, LG7, MG10, and LG10, which display the lowest levels. Intermediate levels are observed in genotypes like DG4, MG2, and LG3. Significant variation (F2, 267 = 61.47, *p* < 0.001) also exists among deep green, middle green, and light green groups (Figure 4H). These findings highlight the diverse carotenoid profiles within sesame genotypes, offering insights into their photosynthetic pigment composition.

#### 2.5.2. Variation in Chlorophyll Concentration across the Different Leaf Disk Localization along the Leaf Blade

Regarding the leaf disk localization, the results of ANOVA analyses highlight the substantial variations in Chla+b among leaf disk positions (F2, 267 = 4.71, *p* < 0.001), providing valuable insights into the diversity of chlorophyll production within the plant specimens studied. Figure 5 presented the outcomes of the mean comparisons for the leaf disk positions of tip, middle, and base along the leaf blade of sesame genotypes. Specifically, the “tip” and “middle” leaf disk localizations (mean = 31.05 and 29.14, respectively) share the same letter, indicating that their chlorophyll content does not significantly differ. Conversely, both the “tip” and “middle” positions exhibit significantly higher chlorophyll content compared to the “base” leaf disk localization (mean: 27.35).

### 2.6. Chloroplast Anatomical Variation between Deep Green and Light Green Sesame Leaves

The electron micrographs of chloroplasts obtained from the mesophyll tissue of leaves of both deep green and light green sesame genotypes (Figure 6A) reveal notable differences in their chloroplast structures, encompassing grana, thylakoids, and lipid droplets. The deep green leaves exhibit a higher density of chloroplasts within their mesophyll cells, as presented in Figure 6(Bb), which typically translates to greater efficiency in capturing and converting light energy into chemical energy. In contrast, the light green leaves display reduced chloroplast densities (Figure 6(Cb)). Likewise, in the deep green leaves, there was greater density and a more well defined stacking of grana (Figure 7(Cc)), enabling the more effective capturing of light and more effective energy transfer during photosynthesis. Conversely, the light green leaves featured fewer and less tightly stacked grana (Figure 6(Bc)), potentially resulting in less photosynthetic efficiency. Furthermore, the deep green leaves manifest a higher abundance of thylakoids, while light green leaves exhibit fewer thylakoids. Additionally, the deep green leaves were characterized by larger and more numerous lipid droplets, indicative of their enhanced energy storage capacity, while, in contrast, the light green leaves have smaller or fewer lipid droplets (Figure 6(Cc)), suggesting a reduced capacity for energy storage.

## 3. Discussion

This study identified significant variations in chlorophyll content, assessed through the SVI, and the chlorophyll/carotenoid concentrations per leaf area, across diverse factors such as genotype diversity, greenness categories, and leaf disk positions. The findings revealed strong associations between SVI, chlorophyll ratios, total chlorophyll, carotenoid concentrations, and genotype variation, indicating distinct sesame genotypes with varying chlorophyll levels. These variations likely stem from genetic differences influencing chlorophyll production and synthesis pathways. Understanding genotype-specific chlorophyll variations is crucial for selecting genotypes aligned with specific objectives, such as a higher chlorophyll extraction or other desired traits. Furthermore, the categorization of sesame genotypes into DG, MG, and LG groups based on their greenness showed the substantial influence of visual leaf appearance on chlorophyll content, measured as SVI, chlorophyll, and carotenoid concentration. DG plants showed the highest chlorophyll content, MD plants exhibited an intermediate chlorophyll content, and LG plants showed the lowest chlorophyll content. This variation in significance suggests that that DG plants have a significantly higher chlorophyll content on average, followed by the MG category, while the LG category has the lowest mean chlorophyll content, indicating that plants in this category exhibit significantly lower chlorophyll content values compared to the other two groups. This variation has practical implications: rapid estimates of chlorophyll content can be made by visually assessing leaf greenness. This could be particularly advantageous for applications in large-scale agriculture and plant breeding initiatives, eliminating the need for sophisticated equipment. Additionally, the study indicated further variance within the DG, MG, and LG categories. For example, some genotypes within the deep green category exhibited a higher chlorophyll content than others. This variation may be attributed to genetic factors [32], environmental conditions [3,33], or specific growth-related elements associated with each genotype [34].

Interestingly, the study found a similar pattern between the concentration of carotenoids and chlorophyll across all factors, including genotypes, greenness categories, and leaf disk positions. This alignment with previous research, which established a positive correlation between total carotenoids and chlorophyll in cauliflower [35], suggests a consistent relationship between these pigments. In the context of sesame genotypes grown in warm and tropical regions with high temperatures, the higher levels of carotenoids in deep green sesame genotypes compared to light ones can be attributed to the plant’s adaptation to its environment, characterized by frequent abundant, high-intensity sunlight [36]. Studies indicated that carotenoids serve as a photo-protective mechanism, dissipating excess light energy as heat, thus safeguarding the photosynthetic apparatus [36,37]. Consequently, deep green sesame genotypes, which were characterized by higher chlorophyll concentrations in this study, may have evolved to excel in these conditions by producing more carotenoids. Moreover, high temperatures can lead to oxidative stress in plants due to the increased production of reactive oxygen species (ROS), and carotenoids were reported to possess antioxidant properties, enabling them to effectively mitigate this stress [37,38]. This suggests that deep green sesame genotypes, with their higher carotenoid levels, could be better equipped to cope with temperature-related stress. The observed correlation between carotenoid and chlorophyll concentrations in sesame leaves in general suggests that sesame plants in warm and tropical regions could adapt to their environment by producing greater amounts of carotenoids. These carotenoids serve dual purposes, acting as both photo-protective agents and antioxidants, ultimately contributing to the plant’s resilience in high-temperature and high-light conditions.

Regarding the leaf disk localization, the SVI and the pigment concentration measurements reveal substantial variations across different leaf disk localizations within the sesame leaves. These results indicate a clear and distinct chlorophyll gradient along the leaf blade. These findings underscore the critical importance of considering leaf disk localization when assessing chlorophyll content in sesame genotypes, as this profoundly affects the recorded measurements. Our results are consistent with previous studies in various crops, including wheat [39], maize [40,41] and rice [41,42], that indicated that chlorophyll was not evenly distributed along the leaf blade. In this study, specifically, the leaf tip exhibited the highest chlorophyll content, followed by the 75.00% localization and the middle localization, while the 50% and 25.00% localizations displayed an intermediate chlorophyll content, and the leaf base showed the lowest chlorophyll content. The observed variations in chlorophyll content can be attributed to several factors, including differences in light exposure, nutrient distribution, leaf age, and physiological factors [34,43,44,45]. The leaf tip may receive more direct sunlight, which stimulates greater chlorophyll production for photosynthesis compared to other positions, while the relatively older portion nearer to the base tended to have a lower chlorophyll content due to the reduced chlorophyll production with leaf maturity [46]. Variations may also arise from differences in nutrient distribution within the leaf, with the base receiving fewer nutrients, impacting chlorophyll production [34]. Furthermore, the leaf’s physiological characteristics, such as its ability to retain chlorophyll or respond to environmental stress, can vary along the blade, contributing to differences in chlorophyll content [34,46]. The study’s key conclusion in this regard is, therefore, the existence of a significant chlorophyll gradient along the sesame leaf blade, with the leaf tip to the 75% portion of the leaf blade having the highest chlorophyll content, and the 25% portion to the base having the lowest. This gradient has implications for various applications, including understanding the leaf’s photosynthetic efficiency and identifying the optimal localization for chlorophyll extraction. To ensure accurate and consistent results in chlorophyll content analyses, it is vital to consider leaf disk localization and establish standardized protocols for localization when conducting measurements, minimizing variations caused by leaf position. For applications requiring a higher level of chlorophyll extraction, focusing on leaf tips and the 75% position is recommended due to their consistently higher chlorophyll content.

The results obtained from the chlorophyll extraction experiment, which involved a comparison of various solvents and procedures (Figure 7A–C), showed significant variations in Chla, Chlb, and total chlorophyll content. These distinctions underscore the substantial impact of both solvent choice and extraction methodology on the efficiency of chlorophyll extraction (Figure 7B). Specifically, the most remarkable outcomes are achieved with DMSO incubated for 25 min at 65 °C (DMSO) and 96% ethanol incubated, for 10 min at 85 °C (96%E85), which extracted significantly higher levels of Chla, Chlb, and total chlorophyll content. It is worth noting that the first extraction in both solvents proved to be the most effective, making a significant contribution to the total chlorophyll content (Figure 7C). These findings are in accordance with previous research, which has consistently demonstrated that DMSO and 96% ethanol are highly efficient solvents for chlorophyll extraction [12,24,31]. Previous investigations have explored several methods for evaluating chlorophyll content in higher plant leaves, and several studies have substantiated that chlorophylls, which are lipid-soluble compounds found in plant tissues, can be effectively extracted using water-miscible organic solvents [4,16,19,20,21,22,23,24,25,26,27].

However, consideration of various factors, including cost-effectiveness, time, safety, and environmental regulations in the laboratory or research setting, is also crucial when choosing an appropriate solvent. In this context, ethanol emerges as a practical and efficient choice for routine chlorophyll extraction, owing to its safety, cost-effectiveness, availability, and user-friendliness [26,47,48,49]. This conclusion is reinforced by prior research that has highlighted the toxicity and noxious odors associated with the DMSO method [29] and the sensitivity of DMSO extraction to temperature fluctuations [30]. Additionally, ethanol has been reported to be a less harmful yet efficient extraction solvent [31]. It is important to note that that while both solvents resulted in comparable results, the use of 96% ethanol incubated for 10 min at 85 °C (96%E85) simplifies the process and ensures consistent results across experiments. Nevertheless, researchers should always evaluate the specific requirements of their experiments and adhere to their lab safety protocols when making a final decision.

The examination of leaf anatomy, specifically the internal structure of chloroplasts within sesame mesophyll tissue leaves, provides valuable insights into their functions and adaptations. In this study, we investigate the anatomical variations between deep green and light green sesame leaves, providing insight into how these differences impact chlorophyll content and distribution. Our examination revealed significant anatomical variations in the internal structure of the leaves between deep green and light green sesame plants, primarily linked to chloroplast density and pigment-producing structures. The deep green sesame leaves exhibited high chloroplast density, well-organized grana, abundant thylakoids, and numerous lipid droplets, all of which point to efficient light capture and energy transfer during photosynthesis. Conversely, the light green leaves displayed fewer and less densely stacked grana, reduced thylakoids, and smaller lipid droplets, indicating lower photosynthetic efficiency. These findings support earlier research, underlining that deep green leaves excel in converting light energy into chemical energy, primarily due to their high chloroplast density [50,51,52,53]. On the other hand, the light green leaves, which were characterized by a lower chloroplast density, may be adapted to high-light environments with strategies to reduce the risk of photo-damage. They feature a loosely arranged mesophyll cell structure that enhances air circulation, reducing the likelihood of overheating [51]. This adaptation regulates the amount of absorbed light. As a result, the variations in leaf anatomy [54] and tissue structure [55] can influence pigment extraction from plants. This underscores the importance of using appropriate extraction procedures and solvents to achieve optimal chlorophyll extraction in plants. Therefore, this knowledge is crucial in chlorophyll extraction studies, facilitating the optimization of techniques for accurate chlorophyll quantification.

## 4. Materials and Methods

### 4.1. Selection of Sesame Genotypes and Pigment Groups

For this study, a total of 30 sesame (*Sesamum indicum* L.) genotypes were chosen, representing three distinct pigment groups based on leaf coloration: light green (LG), middle green (MG), and deep green (DG). Each pigment group comprised 10 genotypes, carefully selected from a larger pool of sesame accessions representing various geographical areas worldwide. These genotypes were cultivated in field experiments using a randomized complete block design (RCBD) with three replications. The selected genotypes formed the core of our investigation, which aimed to optimize SPAD value index measurements, quantify pigment concentrations, explore different extraction solvents, and determine the most suitable leaf disk positions for efficient chlorophyll extraction and quantification.

### 4.2. Measurement of SPAD Value Index (SVI)

To assess the SVI, we identified five plants situated in the middle of the second row for each genotype, which were planted in the RCBD field layouts. The selected plants were labeled and subjected to Soil Plant Analysis Development (SPAD) value assessment using a SPAD meter (SPAD-502, Konica-Minolta, Osaka, Japan). To ensure the precision of our readings, we calculated the average of five measurements taken along each leaf blade, as marked in Figure 8. It is important to note that we specifically focused on fully extended young leaves of similar size, all originating from the main stem of the five selected plants at their 50% flowering stage, in order to obtain accurate measurements of photosynthetic pigments.

### 4.3. Comparative Chlorophyll Extraction Procedures

After measuring the SPAD value indices of the leaves, we conducted a comprehensive comparative analysis that encompassed six distinct chlorophyll extraction methods. These methods included both heated and cold assay techniques.

The heated assay procedures comprised the following five extraction techniques:75% ethanol (75%ETHO) at 65 °C for 25 min: Leaves were immersed in 75% ethanol and incubated at a temperature of 65 °C for 25 min.Boiling in hot water followed by extraction with 96% ethanol (96%ETHO) at 85 °C for 3 min: Initially, leaves were boiled in hot water, after which they were extracted using 96% ethanol at a temperature of 85 °C for 3 min.96% ethanol (96%ETHO) at 65 °C for 25 min: Leaves were directly subjected to extraction using 96% ethanol at a temperature of 65 °C for 25 min.96% ethanol (96%ETHO) at 85 °C for 10 min: Leaves were directly immersed in 96% ethanol and incubated at a temperature of 85 °C for a duration of 10 min.Dimethyl sulfoxide (DMSO) at 65 °C for 25 min: We employed a method utilizing Dimethyl sulfoxide (DMSO) for extraction, with incubation at a temperature of 65 °C for 25 min.

Conversely, the cold assay approach involved the utilization of 96% ethanol (96%ETHO) extraction at a temperature of 4 °C, with an extended incubation period of 24 h.

To separately determine the content of Cha and Chb, fully extended young leaves of three plants in the middle from the second row of each genotype were detached from the plant to extract chlorophylls using a modified method described by [4]. The contents of chlorophyll a and chlorophyll b are expressed as microgram (µg) of chlorophyll per leaf area (cm^2^). For each extraction method used in this study, five circular leaf disks, each 15 mm in diameter, were punched from the leaf portion for which the SPAD value indices were measured using a cork borer. Subsequently, the leaves were rapidly frozen in liquid nitrogen and stored at −80 °C in a deep freezer until the extraction procedure commenced. During the heated assay techniques, the leaf disks were removed from the deep freezer and promptly placed into 8 mL tubes. To each tube, 5 mL of the respective organic solvent was immediately added. The tubes were then incubated in a hot water bath at the specified temperature for the designated duration. In the boiling method, leaves were briefly dipped into boiling water for 10 s and placed on filter paper to remove excess water, after which the organic solvents were added. For the cold assay techniques, following the immediate addition of 5 mL of the organic solvent (96% ethanol), each tube was wrapped in aluminum foil to shield it from light and then incubated at 4 °C in the dark for 24 h.

### 4.4. Estimation of Chlorophyll Content

After the incubation period, spectrophotometer measurements were made by transferring 300 µL of sample into a 1 cm pathlength quartz cell and reading absorbance in a spectrophotometer with a resolution of 1 nm bandwidth (UV-2550, Shimadzu, Kyoto, Japan). Light absorbance measurements were made at 665 nm, 649 nm, and 470 nm, corresponding to the maximum absorption of chlorophyll a (Chla), chlorophyll b (Chlb), chlorophyll total (Chla+b), and carotenoids (Chlx+c), respectively. We employed the updated determined extinction coefficients and equations from [3,5,56] to calculate specific pigment content with respect to the type of solvent used. The represented pigment contents were expressed in units of μg (cm^2^)⁻^1^, representing the amount of chlorophyll per unit area of the leaf.

For the ethanol 96% (*v*/*v*) solvent, the concentrations of chlorophyll a, chlorophyll b, total chlorophyll, and carotenoids were calculated using the following equations:

Ethanol 96% solvent [3,5]:Chla = [13.95 × A665 − 6.88 × A649](1)
Chlb = [24.96 × A649 − 7.32 × A665](2)
Chla+b = [6.63 × A665 + 18.08 × A649](3)
Chlx+c = (1000 × A470 − 2.05 × Chla − 114.8 × Chlb)/245(4)

For the dimethyl sulphoxide (DMSO) solvent [56]:Chla = [12.47 × A665 − 3.62 × A649] (5)
Chlb = [25.06 × A649 − 6.5 × A665](6)
Chl_x+c_ = (1000 × A480 − 1.29 × Chla − 53.78 × Chlb)/220(7)
where A is the absorption at the referenced wavelength and chlorophylls a and b are summed to obtain the total chlorophyll concentration. These analyses allowed us to accurately determine the chlorophyll content in the leaves for each extraction method and solvent type used.

### 4.5. Leaf Disk Localization Determination Procedures

After selecting the appropriate solvent and optimizing the chlorophyll extraction methods, we proceeded to determine the positions of leaf disks. This was accomplished using a cork borer with a 15 mm diameter. We adhered to the criteria mentioned earlier, choosing young, fully extended leaves without any visible damage or signs of disease from the tagged plants with the measured SPAD value indices. Each selected leaf was divided into five equal sections using a ruler, and the positions were marked as follows: tip, 25%, 50%, 75%, and base (refer to Figure 8). This marking was made before detaching the leaf from the plant. Beginning with one leaf, we carefully positioned the core borer over the marked point along the leaf blade, which was placed on a clean, flat cutting board surface. The cork borer was then pressed firmly and twisted slightly to create a clean circular leaf disk. Subsequently, the leaf disk was removed and placed into a labeled tube. It was then rapidly frozen using liquid nitrogen and stored at −80 °C in a deep freezer until the extraction procedure was initiated. This leaf disk punching process was repeated for each marked position on every leaf from the tagged plants within the 30 genotypes. These genotypes were representative of three groups of sesame genotypes: deep green, middle green, and light green. Once all the leaf disks had been punched, properly labeled, and stored in the deep freezer, we proceeded with the chlorophyll extraction procedures using the recommended solvent, (96% ethanol incubated at 85 °C for 10 min, following the methods described in the previous sections.

### 4.6. Collection and Preparation of Sesame Samples for Electron Microscopy

Mature leaves from both deep green and light green sesame genotypes were carefully selected for sampling, employing clean and sterilized tools to minimize contamination risks. To ensure a representative sample set, multiple leaves were collected from various plants of each genotype. During sample collection, great care was taken to handle the leaves gently, minimizing any potential damage. Following collection, the leaves were rinsed with distilled water to remove any accumulated dust and debris. Subsequently, excess moisture was gently blotted from the leaves using clean paper towels. From each leaf, small sections measuring approximately 1 cm² were then carefully excised using a sharp blade. These leaf sections were then placed into labeled sample containers and stored in a cool, dry environment, shielded from direct sunlight. The samples were maintained at a consistent temperature of 4 °C to prevent any degradation of their structural integrity until they were ready to be sent for electron microscopy analysis. For the microscopic analysis of leaf morphology, Transmission Electron Microscopy (FEI Tecnai 20 Transmission Electron Microscopy) was utilized to provide high-resolution imaging of the cellular structures within the sesame leaf samples.

### 4.7. Statistical Analysis

In this study, we utilized an analysis of variance (ANOVA) to compare the mean chlorophyll values obtained using different extraction methods, assess differences in SPAD values among the 30 sesame genotypes selected for the study, examine variations in SPAD values among sesame genotypes grouped by leaf coloration, and evaluate disparities in chlorophyll content based on the position of the leaf disks sampled for analysis. Additionally, we employed the Student–Newman–Keuls (SNK) test for post-hoc analysis to conduct pairwise comparisons between group means. This allowed us to identify specific extraction methods, genotypes, greenness groups, or leaf disk positions that exhibited statistically significant differences in chlorophyll content.

## 5. Conclusions

This study offers valuable insights into the optimization of chlorophyll extraction techniques and the factors that influence chlorophyll content in sesame leaves. It underscores the significance of factors such as genotype-specific variations, greenness categories, and leaf disk positions in determining chlorophyll levels. The study places strong emphasis on the careful selection of appropriate solvents for chlorophyll extraction. The findings of this study suggest considering the pros and cons of various extraction solvents, including aspects like safety, cost, and compliance with environmental regulations. Remarkably, for routine chlorophyll extraction, ethanol stands out as a recommended choice due to its practicality and efficiency. Additionally, for applications necessitating higher chlorophyll extraction, a focus on leaf tips and the 75% position along the sesame leaf blade is recommended. These positions consistently produce a higher chlorophyll content and are preferable for precise chlorophyll quantification. Maintaining consistency in positioning can minimize variations attributed to leaf position, leading to results that are more reliable. Furthermore, the examination of leaf anatomy highlights notable distinctions between deep green and light green sesame leaves, underscoring the importance of employing suitable extraction procedures and solvents. This knowledge is instrumental in optimizing techniques for chlorophyll extraction studies, ultimately contributing to precise chlorophyll quantification and facilitating applications across various fields.

## Figures and Tables

**Figure 1 plants-13-01589-f001:**
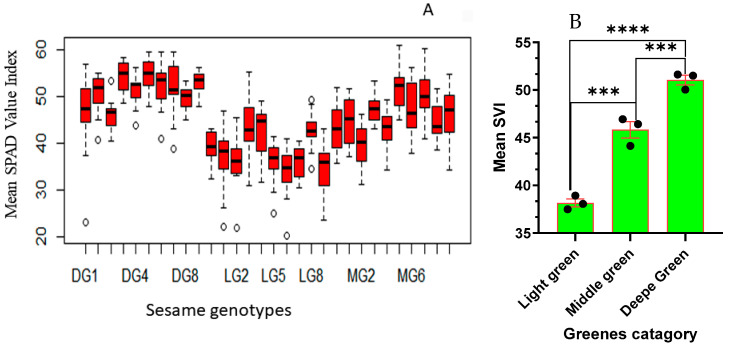
Assessment of SPAD Value Indices for: (**A**) Different sesame genotypes (genotypes are arranged in ascending order from left to right: DG1, DG10, DG2–DG9; LG1, LG10, LG2–LG9; MG1, MG10, MG2–MG9); (**B**) Categories of sesame greenness groups. The bars represent the mean values, and error bars depict one standard error (SE) pair comparison of the *t*-test. The experiment involved 10 genotypes for each category, each replicated three times. Significant level represents, “***” *p* ≤ 0.0001, “****” *p* ≤ 0.00001.

**Figure 2 plants-13-01589-f002:**
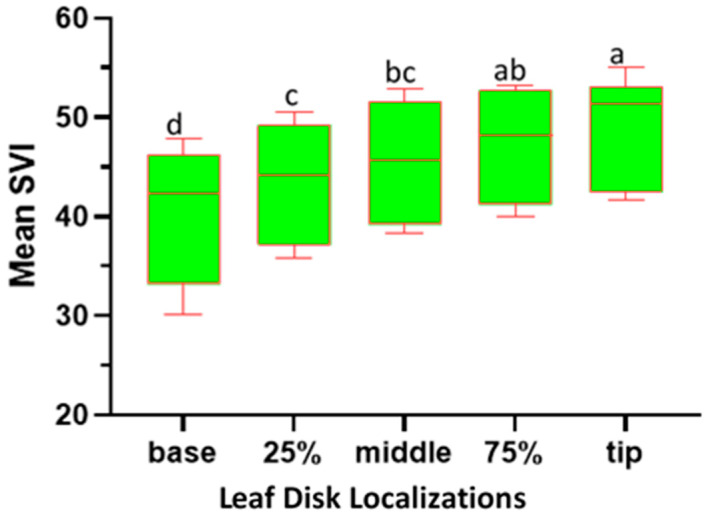
SVI variability across various leaf disk localizations alongside of the sesame leaf blades. Box plots with the same letter(s) indicate no significant difference, as per the SNK test, at a 5% significance level. (Each box represents the Interquartile Range (IQR); the line inside the box (median line) represents the median value; the whiskers extending from the box represent the minimum and maximum values within 1.5 times IQR).

**Figure 3 plants-13-01589-f003:**
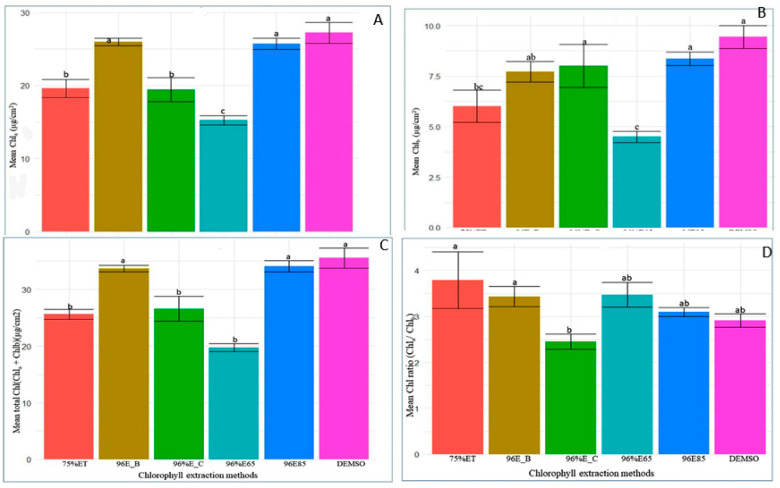
Comparative efficiency of various chlorophyll extraction methods for (**A**) Chla, (**B**) Chlb, (**C**) Chla+b, and (**D**) Chla/b. Extraction methods include DMSO (DMSO), 96%E85 (96% ethanol, 10 min at 85 °C), 96%E_B (boiled leaf in 96% ethanol, 3 min at 85 °C), 96%E_C (96% ethanol, 24 h at 40 °C, cold method), 75%ET (75% ethanol, 25 min at 65 °C), and 96%E65 (96% ethanol, 25 min at 65 °C). Bar plots with the same letter(s) indicate no significant difference as per the SNK test at a 5% significance level. The bars represent the mean values and error bars depict one standard error (SE) from ANOVA. The experiment involved 30 genotypes, each replicated three times.

**Figure 4 plants-13-01589-f004:**
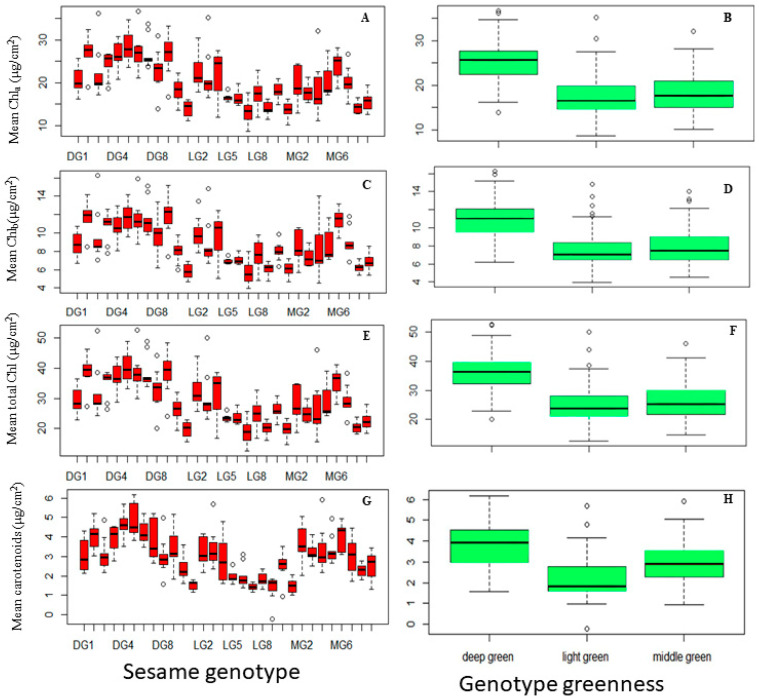
Variation in pigment concentrations in sesame leaves across different genotypes and greenness categories: (**A**) chlorophyll a (Chla), (**B**) chlorophyll b (Chlb), (**C**) total chlorophyll (Chla+b), and (**D**) carotenoid concentration; greenness categories: (**E**) chlorophyll a (Chla), (**F**) chlorophyll b (Chlb), (**G**) total chlorophyll (Chla+b), (**H**) carotenoid concentration. Sesame genotypes (arranged from left to right in ascending order): DG1, DG10, DG2–DG9; LG1, LG10, LG2–LG9; MG1, MG10, MG2–MG90. Each box represents the Interquartile Range (IQR); the line inside the box (median line) represents the median value; the whiskers extending from the box represent the minimum and maximum values within 1.5 times IQR; individual data points beyond 1.5 times IQR in the light green and middle green represent outliers.

**Figure 5 plants-13-01589-f005:**
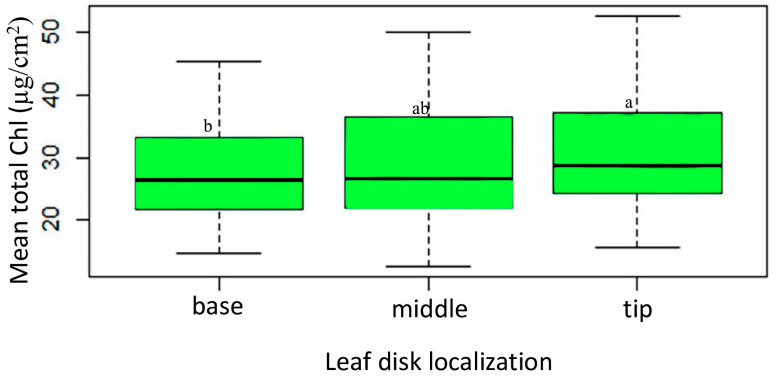
Total chlorophyll (Chla+b) content variation among leaf disk localization along across sesame genotypes’ leaf blades. (Each box represents the Interquartile Range (IQR); the line inside the box (median line) represents the median value; the whiskers extending from the box represent the minimum and maximum values within 1.5 times IQR). The boxs followed by the same letter (s) are not significantly different using SNK test at a 5% level of significance.

**Figure 6 plants-13-01589-f006:**
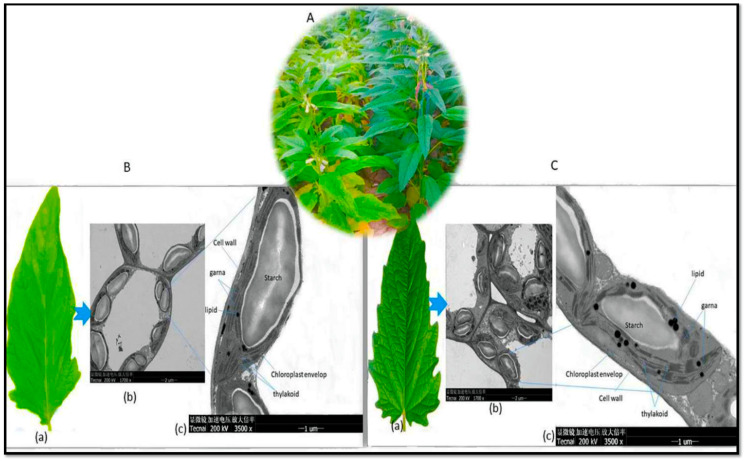
Electron micrographs of chloroplasts within the mesophyll tissue: (**A**) chloroplasts from light green (left) and deep green (right) sesame genotypes in the field; (**B**) (**a**) a single leaf from the light green sesame genotype; (**b**) multiple chloroplasts; (**c**) a thin section of a single chloroplast, revealing the cell wall, chloroplast envelope, starch granules, lipid (fat) droplets, grana (stack of thylakoids), and thylakoids; (**C**) (**a**) a single leaf from the deep green sesame genotype; (**b**) multiple chloroplasts; (**c**) a thin section of a single chloroplast, showing the cell wall, chloroplast envelope, starch granules, lipid (fat) droplets, grana (stack of thylakoids), and thylakoids.

**Figure 7 plants-13-01589-f007:**
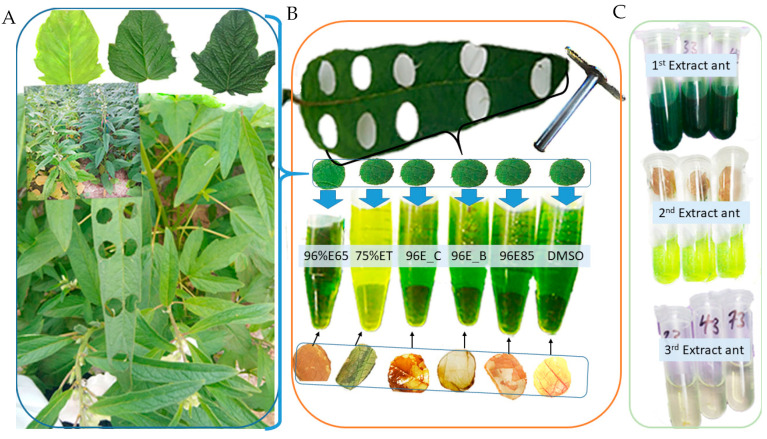
Comparative analysis of extraction procedures: (**A**) sesame leaves of different genotypes; (**B**) various extraction techniques with leaf disk positions, perforations, and leaf disk colors before and after extraction for each method. Extraction methods included DMSO (DMSO), 96%E85 (96% ethanol, 10 min at 85 °C), 96%E_B (boiled leaf in 96% ethanol, for 3 min at 85 °C), 96%E_C (96% ethanol, for 24 h at 40 °C; the cold method), 75%ET (75% ethanol for 25 min at 65 °C), and 96%E65 (96% ethanol for 25 min at 65 °C); (**C**) extractant yield from the first, second, and third extraction cycles using 96% ethanol incubated at 85 °C for 10 min.

**Figure 8 plants-13-01589-f008:**
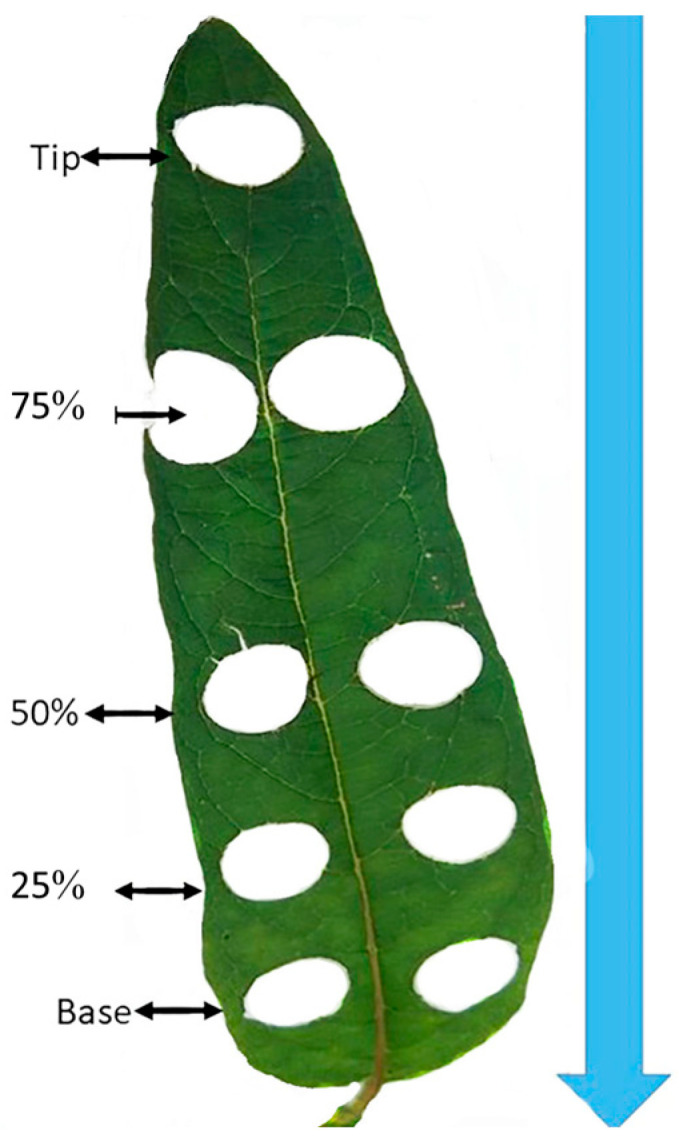
Leaf disk position marks.

**Table 1 plants-13-01589-t001:** Mean ± SE (µg/cm^2^) extraction efficiency of DMSO and 96% ethanol across three extraction frequencies.

Variable	Solvent	Extraction Frequency	Mean ± SE (µg/cm^2^)	% of Extraction Efficiency	Min.	Max.
Chla	96% Ethanol	96%ETOH extract-1	23.82 ± 073 c	81.05	20.12	25.45
		96%ETOH_extract-2	4.97 ± 0.31 de	16.92	3.57	6.18
		96%ETOH extract-3	0.60 ± 0.67 e	2.03	0.27	0.82
		96%ETOH total	29.39 ± 0.82 ab	100.00	25.67	31.76
	DEMSO	DEMSO_extract-1	27.22 ± 1.42 bc	81.04	19.96	30.84
		DEMSO extract-2	5.83 ± 0.98 d	17.36	1.83	9.89
		DEMSO extract-3	0.54 ± 0.09 e	1.62	0.35	0.98
		DEMSO_total	33.59 ± 2.20 a	100.00	24.54	40.79
	F(7, 48)		172.40			
	*p*		<0.001			
Chlb	96% Ethanol	96%ETOH extract-1	11.09 ± 0.78 ab	84.98	8.72	14.98
		96%ETOH_extract-2	1.73 ± 0.12 c	13.24	1.18	2.10
		96%ETOH extract-3	0.23 ± 0.03 c	1.78	0.07	0.34
		96%ETOH total	13.05 ± 0.83 a	100.00	10.43	17.19
	DEMSO	DEMSO_extract-1	9.44 ± 0.56 b	76.79	7.33	11.44
		DEMSO extract-2	2.47 ± 0.42 c	20.05	0.76	4.16
		DEMSO extract-3	0.39 ± 0.05 c	3.16	0.13	0.56
		DEMSO_total	12.29 ± 0.86 a	100.00	9.82	15.19
	F(7, 48)		98.85			
	*p*		<0.001			
Chla+b	96% Ethanol	96%ETOH extract-1	35.37 ± 1.43 b	82.24	29.64	40.93
		96%ETOH extract-2	6.80 ± 0.43 cd	15.80	4.82	8.40
		96%ETOH_extract-3	0.84 ± 0.10 d	1.95	0.34	1.13
		96%ETOH total	43.01 ± 1.60 a	100.00	37.21	49.57
	DEMSO	DEMSO_extract-1	35.55 ± 1.77 b	79.87	26.92	39.66
		DEMSO extract-2	8.06 ± 1.36 c	18.11	2.52	13.64
		DEMSO extract-3	0.91 ± 0.13 d	2.04	0.57	1.41
		DEMSO_total	44.51 ± 2.89 a	100.00	33.37	54.11
	F(7, 48)		165.71			
	*p*		< 0.001			
Chla/b	96% Ethanol	96%ETOH_extract-1	2.19 ± 0.10 bc	28.07	1.70	2.57
		96%ETOH_extract-2	2.89 ± 0.06 b	37.12	2.61	3.05
		96%ETOH_extract-3	2.71 ± 0.20 b	34.81	2.15	3.84
		96%ETOH_total	7.79 ± 0.25 a	100.00	6.84	8.89
	DEMSO	DEMSO_extract-1	2.91 ± 0.15 b	42.35	2.57	3.46
		DEMSO_extract-2	2.38 ± 0.04 bc	34.60	2.25	2.54
		DEMSO_extract-3	1.58 ± 0.35 c	23.05	0.77	3.44
		DEMSO_total	6.87 ± 0.32 a	100.00	5.98	8.45
	F(7, 48)		117.73			
	*p*		<0.001			

The means followed by the same letter(s) in the same column are not significantly different using SNK test at a 5% level of significance.

## Data Availability

Data will be available when requested.

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
