# Peer review of "Examining Chlorophyll Extraction Methods in Sesame Genotypes: Uncovering Leaf Coloration Effects and Anatomy Variations"

_plants, 2024, doi:10.3390/plants13121589_

Round 1

Reviewer 1 Report

Comments and Suggestions for Authors

The study describes a comparison of different methods for extracting chlorophyll from different Sesame genotypes. Just this part of the manuscript is done quite well and in detail. However, this work requires revision. The absence of Figure 1 is immediately noticeable. It is necessary to correctly number the figures. There are no line numbers, making it difficult to make notes on specific locations in the text. 

Abstract. The first sentence causes misunderstanding. Especially "strategically placing leaf disks". It should be more precise, for example, “places on the leaf blade from which leaf discs were cut.” The word “leaf disc position” is not very good. Perhaps “localization” is a better word. The abstract does not say anything about the results of anatomical studies. The abbreviation SPAD appears in the abstract, but nowhere in the text is there an explanation of what it is. You need to decipher the abbreviation, provide an explanation or provide a link to the reference.

Results. The execution of the figures is not equal. Thus, a very large area is allocated to Figure 6, which is completely unnecessary, while a very small area is completely unreasonably allocated to the very informative Figure 7. Small details and signatures are almost impossible to see there. In the legend under Figure 5, the last sentence is not clear, please rewrite it.

Section 2.2. ...Anatomy... It is not clear which tissues of the leaf blade are shown in Figure 7. Epidermis? Spongy parenchyma? Palisade parenchyma? The number of chloroplasts in these cells may depend on this. This must be specified.

Materials and methods. There is little information about sesame genotypes. Are they different varieties or populations from different geographic areas?

Again, we point out that it is necessary to explain what the SPAD indicator is.

Author Response

Dear Reviewer

Thank you for taking the time to review our manuscript, " Examining Chlorophyll Extraction Methods in Sesame Genotypes: Uncovering Leaf Coloration Effects and Anatomy Variations ." We appreciate your thoughtful comments and suggestions, and we have carefully considered each point you raised.

we address each of your comments in turn and explain how we have revised the manuscript accordingly. We hope that these changes address your concerns and improve the quality of the manuscript.

Please see the attachment for detail responses

Reviewer 2 Report

Comments and Suggestions for Authors

The authors comprehended the important topic of sesame cultivation and studied methods of extracting chlorophyll from this plant. They also studied the effects of various factors on the chlorophyll content of sesame leaves. A number of solvents for chlorophyll extraction were studied.

The paper is edited very carefully. I found no methodological errors.

Minor correction is required in numeration of figures in the work and citation in the text of the authors of the publication marked in the bibliography with the number 4.

Author Response

Dear Reviewer

Thank you for taking the time to review our manuscript, " Examining Chlorophyll Extraction Methods in Sesame Genotypes: Uncovering Leaf Coloration Effects and Anatomy Variations ." We appreciate your thoughtful comments and suggestions, and we have carefully considered each point you raised.

Thank you for acknowledging the importance of our study on sesame cultivation and chlorophyll extraction methods. We are pleased that you found our investigation into the effects of different factors on chlorophyll content valuable. We aimed to provide a comprehensive analysis of chlorophyll extraction techniques to contribute to advancements in sesame cultivation practices.

Round 2

Reviewer 1 Report

Comments and Suggestions for Authors

The authors made the necessary changes to the text of the manuscript. However, the issues remained with the numbering of the Figures. It is unacceptable to place Figure number 1 at the end of the manuscript. Figures are numbered according to their occurrence in the text. Please correct the numbering of the Figures. The Figure in Materials and Methods should be number 8.

Author Response

Thank you for your valuable feedback and suggestions.

We have carefully reviewed your comments and have made all the necessary changes to the manuscript. Specifically, we have corrected the numbering of the figures as per your instructions. The figures are now numbered according to their occurrence in the text, with Figure 1 -7 placed appropriately. Additionally, the figure in the Materials and Methods section has been renumbered to Figure 8.

We appreciate your thorough review and hope that the revisions meet your expectations.

Thank you for your time and consideration.